# Spatial and Temporal Distribution of *Aedes aegypti* and *Aedes albopictus* Oviposition on the Coast of Paraná, Brazil, a Recent Area of Dengue Virus Transmission

**DOI:** 10.3390/tropicalmed7090246

**Published:** 2022-09-14

**Authors:** Silvia Jaqueline Pereira de Souza, André de Camargo Guaraldo, Nildimar Alves Honório, Daniel Cardoso Portela Câmara, Natali Mary Sukow, Sarita Terezinha Machado, Claudia Nunes Duarte dos Santos, Magda Clara Vieira da Costa-Ribeiro

**Affiliations:** 1Microbiology, Parasitology and Pathology (PPGMPP), Laboratory of Molecular Parasitology, Department of Basic Pathology Curitiba, Federal University of Paraná (UFPR), Curitiba 81530-900, Brazil; 2Zoology Department, Federal University of Paraná (UFPR), Curitiba 81531-980, Brazil; 3Laboratório de Mosquitos Transmissores de Hematozoários—LATHEMA, Instituto Oswaldo Cruz, Núcleo Operacional Sentinela de Mosquitos Vetores—Nosmove, Fundação Oswaldo Cruz, Fundação Oswaldo Cruz, Rio de Janeiro 21040-900, Brazil; 4Laboratory of Molecular Parasitology, Department of Basic Pathology Curitiba, Curitiba 81530-900, Brazil; 5Paraná State Health Department, 1st Health Regional, Morretes 83350-000, Brazil; 6Molecular Virology Laboratory, Carlos Chagas Institute, Fiocruz, Curitiba 81350-010, Brazil

**Keywords:** *Aedes aegypti*, *Aedes albopictus*, spatial distribution, occurrence, oviposition, meteorological factors

## Abstract

*Aedes aegypti* and *Aedes albopictus* are considered the most important vectors of arboviruses in the world. *Aedes aegypti* is the primary vector of dengue, urban yellow fever, chikungunya and zika in Brazil, and *Ae. albopictus* is considered a potential vector. Distribution patterns and the influence of climatic variables on the oviposition of *Ae. aegypti* and *Ae. albopictus* were evaluated in Morretes, a tourist city in the coastal area of Paraná State, Brazil, which has recently been experiencing cases of dengue fever. Eggs were collected using ovitraps over a period of one year (September 2017 to September 2018) and reared from hatching until the emergence of the adults. Both *Aedes* species were found in anthropized areas with a high human density index. Findings suggest that the monthly average temperature (LRT = 16.65, *p* = 0.001) had significant positive influences on the oviposition of the *Aedes* species. Considering the wide distribution of DENV around the Paraná coast and the presence of *Ae. albopictus* alongside *Ae. aegypti*, studies on natural arbovirus infection patterns and seasonality are recommended in the region.

## 1. Introduction

*Aedes aegypti* and *Aedes albopictus* are sympatric, cosmopolitan and epidemiologically important species involved in transmitting arboviruses such as dengue (DENV), zika (ZIKV), yellow fever (YFV), and chikungunya viruses (CHIKV) [1,2,3]. These species have similar geographical distribution and share microhabitats in some regions such as Florida and Rio de Janeiro, including artificial vessels containing water [4,5]. These mosquitoes share the same breeding sites and compete for food, which in turn favors one species over the other and alters the distribution or abundance of both, as well as other resident mosquito species [5,6,7].

Studies carried out on mapping the global distribution of *Ae. aegypti* and *Ae. albopictus* point to an expansion in the range of habitat suitability for these species and predict that their dissemination will occur despite environmental changes, since the species are adapting to anthropogenic ecological niches using spatial dispersion [8,9,10]. *Aedes aegypti* had its reintroduction in Brazil in 1976 through the port of Salvador [11], and since then it has been present in all Federation Units [12]. It was reported in 1985 in the state of Paraná, in the municipality of Londrina [13]. After 28 years, the presence of the *Ae. aegypti* mosquito was found on the coast of Paraná, and subsequently one of the main dengue epidemics occurred in the region [14]. *Ae. albopictus* was found in Rio de Janeiro in 1986 [15] and arrived in the state of Paraná two years later in the municipality of Arapongas [16]. Its presence has been registered on the coast of the State since 2001 [17].

*Aedes aegypti* and *Ae. albopictus* occur in tropical and subtropical regions, as such environmental characteristics allow reproduction and provide adequate habitat, and most of Brazil is within this climatic zone [8]. The southern region of Brazil differs from other Brazilian regions due to a distinction between seasons and also by the thermal amplitude; it is the coldest region in the country, with snow in some locations during the winter [18]. The coast of Paraná has a humid subtropical climate (mesothermal) according to the Koppen classification [19,20,21], with the average temperature in the cold months being below 18 °C and an average temperature above 22 °C in the summers, concentrating the rains in the summer, despite not having a defined dry season [22,23,24].

The coastal area of the State of Paraná exhibits a reduced thermal amplitude during the year, probably due to the influence of the warm maritime current from Brazil which causes an increase in air humidity [25] with an average rainfall of 2000 mm to 2200 mm per year. Rainfall patterns are irregular in summer and more constant in spring [22,26]. On the other hand, the temperature presents averages above 37 °C in the summer and below 13 °C in the coldest periods [22], constituting characteristics which enable both *Ae. aegypti* and *Ae. albopictus* to proliferate in the region.

The Paraná State Health Department (SESA/PR) reported the first important dengue epidemic in the city of Paranaguá in 2016, as well as the first occurrence of ZIKV and CHIKV infections [14]. Paranaguá and Antonina are the main city ports along the coast of the State of Paraná, Brazil.

All cargoes which leave the port of Antonina are routed through Morretes, which is also an important tourist city, and *Ae. albopictus* was the predominant species until 2014, when *Ae. aegypti* was first noticed. Thus, the aim of this study is to present the spatial and temporal distribution of *Ae. aegypti* and *Ae. albopictus* along a transect in the city of Morretes. As well as to evaluate the influence of climatic variables on these species in a recent area of dengue in southern Brazil, which have been sympatric since 2014.

## 2. Materials and Methods

### 2.1. Study Area

The city of Morretes is located on the coast in Paraná State, has 15,718 inhabitants and is a part of the first Health Region of Paraná State. It is situated on a plain at 25°38′00″ S latitude and 48°34′00″ W longitude, covering an area of 684.580 km^2^ (Figure 1A,B). The city is characterized by a humid tropical climate, with average temperatures above 37 °C in summer and below 13 °C in winter, an undetermined dry season, hot summers, and infrequent frosts [27,28].

### 2.2. Entomological Survey and DENV Dengue Fever Cases

Oviposition traps are used to collect populations of *Ae. aegypti* and *Ae. albopictus* as tools for mosquito surveillance in different studies in Brazil, since this method has high sensitivity in detecting the presence and comparing infestations in different areas [1,29,30,31,32,33]. *Aedes* eggs were collected using ovitraps consisting of a black container with a capacity of 500 mL of type water, using Eucatex^TM^ paddles fixed by clips which allow for oviposition [34].

The ovitraps were placed at 23 points (Figure 1C) along a transect crossing the city of Morretes, covering places with high and low vegetation cover and with high and low circulation of people, in backyards, external commercial places, tire repair shops and garages, as well as schools, with an average spacing of 191.64 m between them. These ovitraps were installed for 4 consecutive days for 13 months, for 24 h, generating a monthly sampling effort of 2208 h of capture. After removal from the collection field, the straws were taken to the Molecular Parasitology Laboratory at the Federal University of Paraná (UFPR), and analyzed in a stereomicroscope for the presence or absence and number of eggs, which were then reared from hatching until adults emerged. Adult mosquitoes were identified according to collection point, sex, and species using the dichotomous key of Consoli and Lourenço-de-Oliveira [3].

The monthly collection was from September 2017 to September 2018, one year after the first notification of an autochthonous cases of dengue in the municipality of Paranaguá by the Paraná State Health Department (SESA/PR). The data on autochthonous cases of dengue fever in the city of Morretes and coastal municipalities from 2014 until 2020 were obtained by SESA [14].

### 2.3. Climate Variables

Monthly data regarding total precipitation, temperature, and relative humidity (maximum, average, and minimum) were obtained from the Meteorological Station APPA (Administração dos Portos de Paranaguá e Antonina) Antonina at 25°44′71.6″ S latitude and 48°34′43.4″ W longitude, located at 17.8 km from the municipality of Morretes [26]. Meteorological information was collected from 2014 to 2018 in order to determine a possible climatic variation in the three years preceding the experiments.

### 2.4. Statistical Analysis

We first contrasted the number of emerged adults of both species and sex within species through a negative binomial linear model followed by estimated marginal means contrasts to check for the hatching patterns of both *Aedes* species. Then, we calculated the adult hatching peak date and its concentration (r) along the year for each species. This allowed us to assess the oviposition patterns, as well as to compare the circular distribution (i.e., temporal) pattern of this data of each species via a Wallraff rank sum, a test based on Kruskal-Wallis χ^2^ [35].

Next, a preliminary assessment was performed comparing the climate of the study period with that of the previous three years to adjust the comparison model between climatic variables and mosquito frequency using linear mixed models (LMM) and considering the month as a random factor. We graphically checked for each model validity through the residual plots against the data to verify homoscedasticity assumptions, normality of residuals, and influential values (i.e., outliers). The post-hoc Tukey’s test, followed by the Bonferroni correction, was used to compare the absolute number of adults between years. The analyses were performed in R 3.4.2 [36], using lme4 [37], emmeans [38], cowplot [39] and ggplot2 [40].

The circular package was used to calculate the concentration (r) and the average oviposition date for each mosquito species frequency, thus assessing the oviposition patterns, as well as to compare the circular distribution (i.e., temporal) pattern of this data from each species via a Wallraff rank sum, a test based on Kruskal-Wallis χ^2^ [35].

Poisson Generalized Linear Mixed Model (GLMM) followed by variable selection through likelihood ratio (LRT) [41] test was used to assess the influence of climatic and environmental factors on the oviposition patterns for both mosquito species in the study area, considering traps and months as random factors in these models. The model was graphically validated, as above. The number of adults was the response variable, while species, monthly average temperature, minimum relative humidity were the predictor variables. All other available climate variables monthly, i.e., rainfall, minimum temperature, maximum temperature, maximum relative humidity were excluded due to collinearity (r > 0.7).

Mosquitoes oviposition was represented using a Kernel cut-off estimator. The Kernel map estimates function values at intermediate points of intervals for point location for an entire area forming a surface whose value is proportional to the intensity of those facing the area [42].

## 3. Results

### 3.1. Spatial and Temporal Distribution of Mosquitoes and DENV Cases

A total of 6462 eggs of *Aedes* species were collected during this study. From these, a total of 1700 (26.30%) mosquitoes were identified from eggs collected over 13 months in Morretes, Paraná. In addition, 1011 (59.47%) of these were identified as *Ae. albopictus*, being 516 (51.03%) males and 495 (48.97%) females; and 689 (40.53%) were identified as *Ae. aegypti*, including 333 (48.33%) males and 356 (51.64%) females. Adult hatching was null for samples taken in all points in August 2018 (Figure 2). The number of hatched mosquito specimens significantly varied (LRT = 25.33, *p* = 0.005) among the remaining months, irrespective of species (LRT = 0.156, *p* = 0.69) and sex (LRT = 0.053, *p* = 0.82). Nevertheless, significant contrasts only occurred between April and November (z = −3.53, *p* = 0.02) and November and December (z = 4.13, *p* = 0.0002).

Considering the 23 collection sites selected, a higher collection of eggs was observed at collection points 15 and 23 (N = 8), being eight months with positive collections), followed by collection points: 9 (N = 7), 17 and 22 (N = 6), 4, 11 and 16 (N = 5), 2, 3, 10, 13, 18 and 19 (N = 4), 1, 5, 8 and 21 (N = 3), and 7 and 20 (N = 2) (Figure 3; Table 1). *Aedes aegypti* identified from eggs were present at different collection points in the central area of Morretes, and the highest numbers were found at collection points 4 (N = 66, 9.57%), 9 (N = 64, 9.28%), 11 (N = 64, 9.28%), 15 (N = 148, 21.48%), 18 (N = 64, 9.28%) and 23 (N = 59, 11.57%) (Figure 3A). *Aedes albopictus* identified from eggs were present in high numbers at collection points: 11 (N = 117, 11.57%), 13 (N = 118, 11.67%), 15 (N = 100, 9.89%), 16 (N = 76, 7.51%), 17 (N = 96, 9.49%) and 23 (N = 127, 12%) (Figure 3B). Only two collection points (12 and 14) showed no evidence of the presence of *Aedes* spp. individuals. Overall, the highest occurrence of *Aedes* spp. was observed in the central area of the city throughout the study period (Figure 3C).

A total of 13 autochthonous dengue fever cases were recorded in Morretes during 2017 (Figure 4), with five of these cases located close to collection points 3, 8, 9, 10 and 18 from this study. Two of these collection points with dengue cases reported (points 3 and 8) only showed the presence of *Ae. albopictus*.

The *Aedes* species identified in this study presented different oviposition patterns (χ^2^ = 209.69, *p* < 0.001). *Aedes albopictus* identified from eggs presented the highest number of adults, with record numbers from October 2017 to March 2018 (r = 0.58) and an average peak between January and February 2018 (Figure 5A). *Aedes aegypti* presented a lower number throughout the year (r = 0.41), with an average peak in mid-February 2018 (r = 0.41) (Figure 5B).

### 3.2. Abiotic Factors

The average temperature during the collection period ranged from 17 °C to 24.9 °C, with the highest average temperature of 24.9 °C from December 2017 to April 2018. The average relative humidity varied from 80% to 87%, with the lowest levels experienced in November 2017 (80%) and the highest in May, June and July 2019 (87.5%). Total precipitation varied from 16 mm (July 2018) to 288 mm (December 2017). Temperature, relative humidity and precipitation levels during the collection period were similar to those recorded the previous three years (Tukey’s HSD; *p* > 0.05) (Figure 6). The average relative humidity varied from 79% to 91% over the previous years, which is not significantly different from the value obtained during the collection period. However, the minimum relative humidity showed different patterns from 2014 to 2015 and from 2015 to 2016 (Figure 6E).

Of the climatic variables assessed, only the average temperature had a significant positive influence on number of hatched adults (LRT = 16.65; *p* < 0.001) (Figure 7), with a slight variation in the total number of adults per trap (σ^2^ = 3.56 ± 1.89) and over the months (σ^2^ = 1.07 ± 1.03). *Aedes* species showed significant differences in the total number of hatched adults per trap, showing higher numbers for *Ae. albopictus* species (ß = 0.51 ± 0.05; z = 9.38; *p* < 0.001).

## 4. Discussion

Some autochthonous cases of dengue were recorded near our collection point during the sampling period. *Aedes albopictus* was the only species identified at collection points 3 and 8, whereas both *Aedes* species were observed at collection points 9, 10 and 18. Collection point 3 was characterized by being less urbanized, and was located close to a high traffic highway; whereas collection points 8, 9 and 10 were located in the central region of the city close to tourist areas, railroads and areas with higher human circulation levels.

The introduction of *Ae. aegypti* in Morretes city was described by the Entomology Laboratory of the Municipality of Morretes from the Health Regional of SESA/PR in 2014 [43]. *Aede**s albopictus* had already been detected in the forest edge in rural habitats of Morretes [17].

Autochthonous dengue fever cases were reported on the coast of Paraná after Paranaguá city reported the presence of *Ae. aegypti* in 2013 [44]. Paranaguá registered its first case in 2014, and two years later Antonina and Morretes. Other cities along the Paraná coast, such as Guaraqueçaba, Matinhos, Pontal do Paraná and Guaratuba, also recorded dengue fever cases in 2016 (Figure 4) [14].

Our results show that both *Ae. aegypti* and *Ae. albopictus* were observed in an urbanized area in the central region of the municipality. The predominance of *Ae. albopictus* found in 61% of the ovitraps can be explained by the geographical characteristics of the municipality being in the Atlantic Forest, which has a strong influence in this urbanized area. *Aedes aegypti* prefer more urbanized environments, which explains the higher numbers in areas of high anthropic influence with movement of large numbers of people.

Recent studies [7,45,46,47] reinforce that *Ae. albopictus* is mainly found in natural oviposition sites [3,32,48]; this preference does not limit the species, as there is a record of its presence in more developed suburban and urban environments, using artificial breeding sites such as plastic containers and tires [33,47,48,49,50]. The coexistence of *Ae. aegypti* and *ae. albopictus* in artificial breeding sites in forested and urbanized areas demonstrates a dominance of *Ae. albopictus* over *Ae. aegypti* under these conditions [5,6,45].

The search for *Ae. albopictus* in artificial places to lay eggs may be correlated with the disorderly growth of urban areas and the lack of waste management, thereby producing changes in the landscape [33,47,48,49,50] in the transition zones between forest and urban areas, which facilitates the dispersal of anthropophilic mosquito species to previously unsuitable habitats [50,51,52,53,54].

Since its arrival in Brazil [15], *Ae. albopictus* has been found in 59% of cities in different regions of the country [55,56]. Moreover, this species has gradually been occupying locations in the urban environment. Although characteristically common in areas with vegetation, it also survives in transitional environments with relatively low vegetation cover and greater dispersion of the human population [5,32,33,49,50]. Later studies have shown that both patterns were identified in Brazil for this mosquito: *Ae. albopictus* was found coexisting with *Ae. aegypti* in the city of Recife, Brazil, in places with urban characteristics where the human population has expanded and in areas with greater vegetation cover [53,57].

The dispersion and domiciliation of *Ae. albopictus* in densely urbanized areas is a concern, since the mosquito competent vector for the transmission of YFV strains [50], related to sylvatic environments, lies in its ability to survive in transition zones [58,59,60]. At the same time, authors have suggested that the circulation of the virus may be occurring among mosquitoes, acting as a possible bridge between wild and urban cycles of the virus [47,57,58,59,60,61]

Of the climatic variables, average temperature presented the highest significant correlation with the species collected. There was also an increase in adult density during months with higher temperatures, such as January and February 2018. This finding was corroborated by the one study carried out in the city of Natal, Brazil, where high temperatures are constant throughout the year, predisposing the region to high *Ae. aegypti* and *Ae. albopictus* infestation; however, its development can be harmed with excess precipitation and relative humidity [57,62,63].

The positive influence of temperature on *Ae. aegypti* proliferation was also recorded in the city of Porto Alegre, State of Rio Grande do Sul, Brazil [64]. A study on the adaptation of *Ae. aegypti* and *Ae. albopictus* pointed out that the increase in temperature favors the species and that temperatures below 13 °C is lethal for both [65]. Custódio et al. [63] demonstrated that temperatures above 24 °C in Campo Grande, Brazil favored the development and temperature of approximately 18 °C prevented the reproduction of *Ae. albopictus.*

Although *Ae. albopictus* has not been identified as a vector for arboviruses in Brazil, monitoring this vector is essential due to its abundant and proven vectorial competence [5,66]. An *A. albopictus* sample infected by DENV and ZIKV was reported in the state of Espírito Santo, but even with epidemiological relevance, it was not possible to identify the infection origin [67]. It is noteworthy that *Ae. albopictus* is a DENV vector in several Asian [68,69,70] Africans countries [45,46], as well as in some temperate countries such as Spain [71,72], France and Italy [73,74,75,76].

*Aedes albopictus* was the predominant species in the city of Morretes until 2014. The first autochthonous cases of DENV and the detection of *Ae. aegypti* had already been reported in Paranaguá in this period.

We suggested that introduction of *Ae. aegypti* was facilitated by the main route that connects Antonina to Paranaguá and passes through the city of Morretes. The tourism, the main economic activity in the city, also contributes to flow of people and means of transport [28].

In conclusion, *Ae. albopictus* is still prevalent in the city of Morretes. The monthly average temperature had significant correlation with the species collected, which in turn had significant positive influences on the oviposition rates of *Ae. aegypti* and *Ae. albopictus*.

Considering that the coast of Paraná has registered autochthonous cases of arbovirus infection [14] mainly accompanied by the arrival of *Ae. aegypti*, it is of utmost importance to understand the seasonality of *Ae. aegypti* and *Ae. albopictus* in this region. In addition, further studies on the presence of *Ae. albopictus* in relation to *Ae. aegypti* and surveillance on natural arbovirus infection are recommended in these areas.

## Figures and Tables

**Figure 1 tropicalmed-07-00246-f001:**
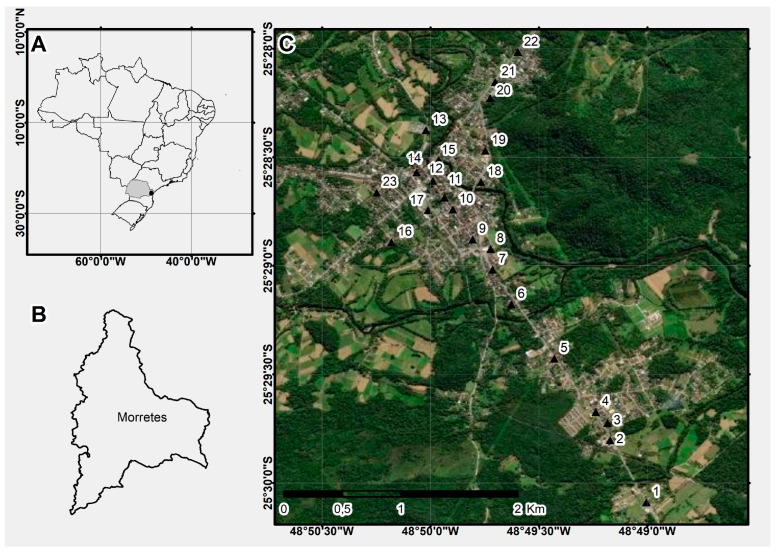
(**A**) State of Paraná with the approximate location of Morretes; (**B**) Municipality of Morretes, Paraná; (**C**) Collection points (1–23).

**Figure 2 tropicalmed-07-00246-f002:**
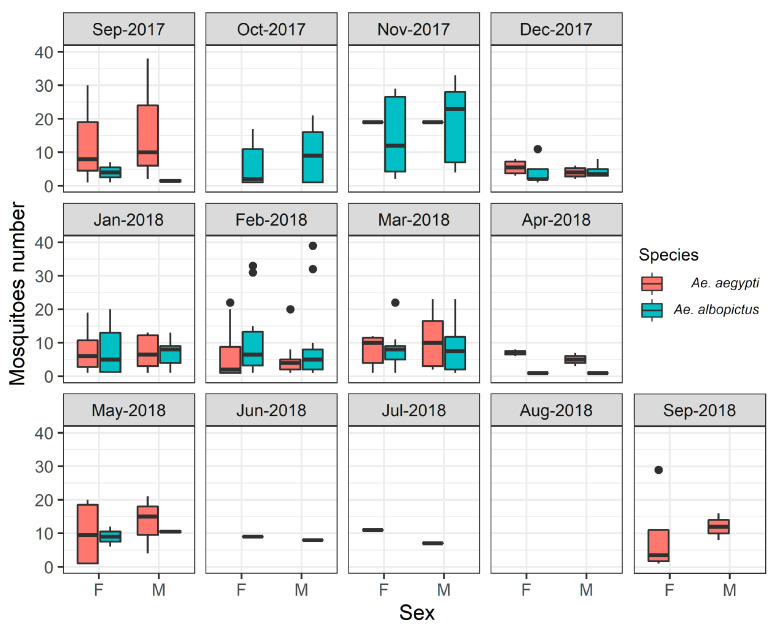
Frequencies of *Aedes aegypti* and *Aedes albopictus* adults by sex identified from hatching eggs, September 2017 to September 2018, Morretes, Paraná, Brazil.

**Figure 3 tropicalmed-07-00246-f003:**
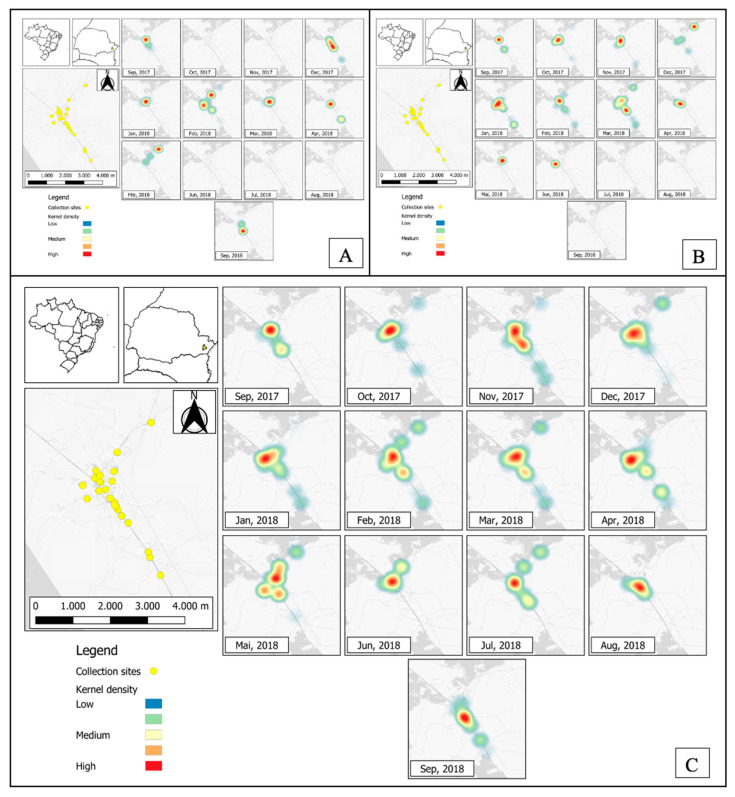
(**A**) Temporal and spatial distribution of *Ae. aegypti* identified from collected eggs; (**B**) Temporal and spatial distribution of *Ae. albopictus* identified from collected eggs; (**C**) Kernel map showing the occurrence of eggs collected in the 23 selected points, Morretes, Paraná, Brazil.

**Figure 4 tropicalmed-07-00246-f004:**
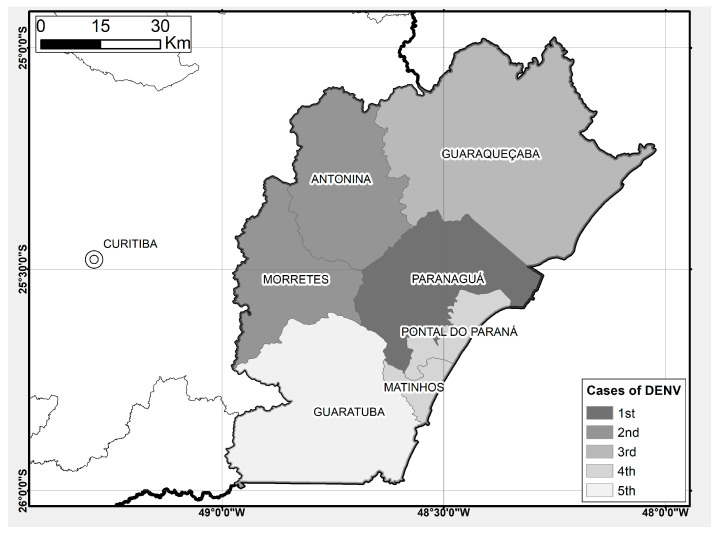
Detection timeline of DENV cases on the coast of the state of Paraná, Brazil.

**Figure 5 tropicalmed-07-00246-f005:**
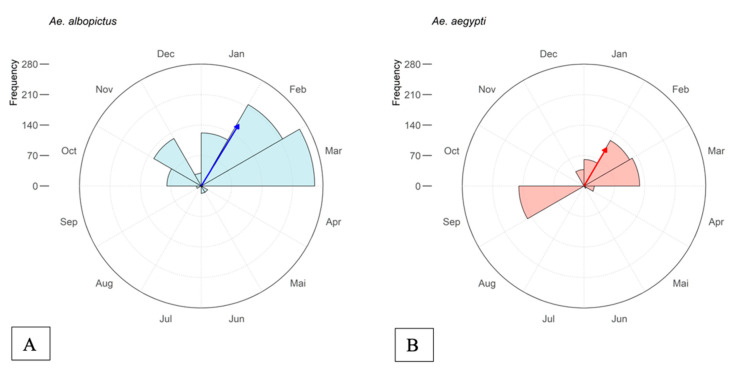
(**A**) Distribution of *Ae. albopictus* identified from eggs collected in ovitraps by month from September/2017 to September/2018 in Morretes, Paraná; (**B**) Distribution of *Ae. aegypti* identified from eggs collected in ovitraps by month from September/2017 to September/2018 in Morretes, Paraná. The vector direction concentration points out the mean peak oviposition date for each species. The longer the vector, the more concentrated the oviposition around the mean peak date.

**Figure 6 tropicalmed-07-00246-f006:**
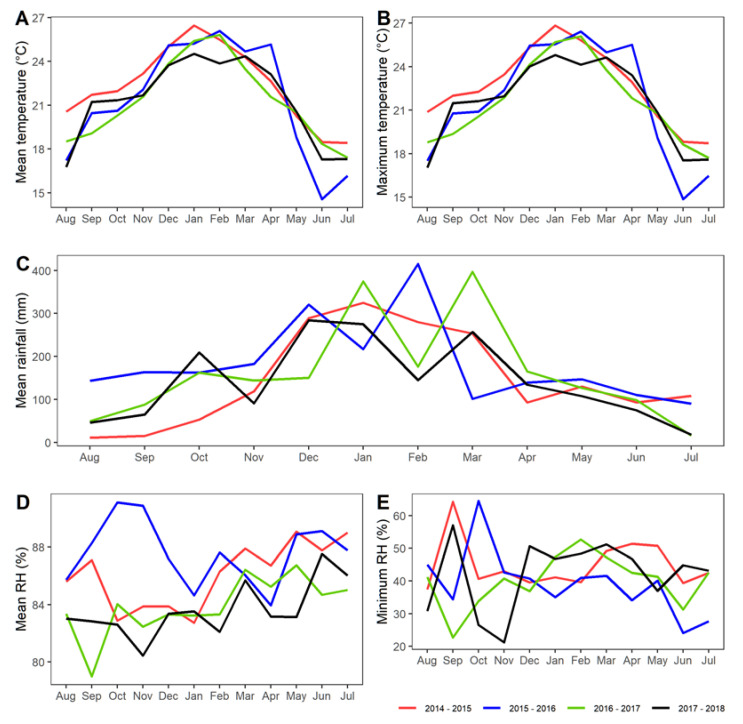
Temporal variation monthly in the average temperature (**A**), average maximum temperatures (**B**), average rainfall (**C**), average relative humidity (**D**), average minimum relative humidity (**E**) in the region of Morretes, Paraná, Brazil. Only the periods of 2014–2015 and 2015–2016 differed from the others regarding the monthly average relative humidity (post-hoc pairwise comparison test; see text).

**Figure 7 tropicalmed-07-00246-f007:**
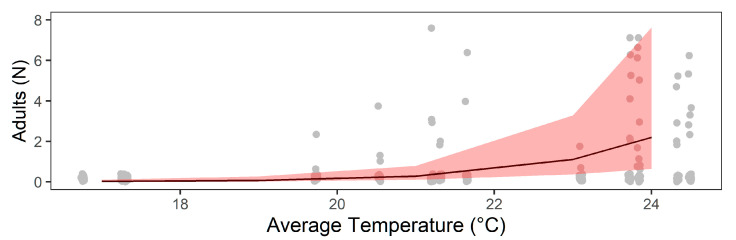
Effect of the average temperature (ß = 1.82 ± 0.42; z = 4.36; *p* < 0.001) on the number of hatched adults *Aedes* from eggs collected in ovitraps between September 2017 and September 2018, Morretes, Paraná, Brazil. The shaded areas are the 95% confidence intervals, and the gray points are the raw data.

**Table 1 tropicalmed-07-00246-t001:** Collections sites in the municipality of Morretes, September 2017 to September 2018, Paraná, Brazil.

Collection Points Positive																								
*Months*	1	2	3	4	5	6	7	8	9	10	11	12	13	14	15	16	17	18	19	20	21	22	23	Total
Sep/17						+			+						+									3
Oct/17		+									+		+								+	+	+	6
Nov/17			+					+	+				+		+	+	+						+	8
Dec/17	+			+	+				+	+					+		+			+		+	+	10
Jan/18		+	+	+					+		+				+	+	+	+	+				+	11
Feb/18	+		+	+	+	+	+			+	+		+		+	+	+	+	+	+	+	+	+	18
Mar/18	+	+	+	+	+	+	+	+	+	+	+		+		+	+	+	+	+			+	+	19
Apr/18		+								+	+					+	+						+	6
May/18															+				+		+	+	+	5
Jun/18																						+		1
Jul/18									+						+									2
Aug/18																								0
Sep/18				+				+	+									+						4
Total	3	4	4	5	3	3	2	3	7	4	5	0	4	0	8	5	6	4	4	2	3	6	8	Positives sites

Legend: + means positives sites.

## Data Availability

Not applicable.

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
