# Peer review of "Spatial and Temporal Distribution of Aedes aegypti and Aedes albopictus Oviposition on the Coast of Paraná, Brazil, a Recent Area of Dengue Virus Transmission"

_tropicalmed, 2022, doi:10.3390/tropicalmed7090246_

Round 1

Reviewer 1 Report

1 - Introduction

- line 59: remove "according to".

- lines 67 and 68: this climatic information must be compatible with that presented in the Material and Methods (lines 80 and 81).

2 - Material and Methods

- line 87: it is also used for Ae. albopictus.

- lines 95 to 97: this sentence needs to be improved.

- lines 98 to 103: this is a very big sentence. It could be divided into two or three small phrases.

3 - Results

- line 158: "were collected" appeared two times in this sentence.

- lines 179 to 182: I think it is Figure 3A and not 2B; the authors did not cite Figure 3B.

- lines 184 and 185: I think it is Figure 3C and not A.

- lines 187 and 188 - Figures 3A and B: These maps (A and B) also represent the spatial distribution, not only the temporal one.

- line 196 - Figure 4: This figure, in my point of view, is not necessary, The authors could comment on the information about dengue in the text and insert neighbor municipalities of Morretes in figure 1.

Author Response

We authors thank the reviewers for their comments and suggestions to improve our manuscript. We have carefully revised the manuscript and considered all modifications as suggested, except for Figure 4. The authors decided to keep it because is clearer about the entry of dengue cases on the coast of Paraná. Please, find below point-by-point responses to reviewer as requested, indicating which changes have been made, and where they have been included in the manuscript. Furthermore, the manuscript was revised by ECB - English Consulting Brazil.

Answers to Reviewer

Reviewer #1:

1. Introduce:

- Line 59: remove "according to".

Authors: Done. It was removed "according to", line 67.

- Lines 67 and 68: this climatic information must be compatible with that presented in the Material and Methods (lines 80 and 81).

Authors: The authors are grateful for this suggestion and this change was adapted according to the methodology, on lines 78 to 79: “On the other hand, the temperature presents averages above 37°C in the summer and below 13°C in the coldest periods [22], constituting characteristics which enable both Ae. aegypti and Ae. albopictus to proliferate in the region.”

2. Material and Methods

- Line 87: it is also used for Ae. albopictus.

Authors: Thank you for pointing this out, the information was adapted, on lines 106 to 108: “Oviposition traps are used to collect populations of Ae. aegypti and Ae. albopictus as tools for mosquito surveillance in different studies in Brazil, since this method has high sensitivity in detecting the presence and comparing infestations in different areas”

- Lines 95 to 97: this sentence needs to be improved.

Authors: This sentence was adapted and placed on lines 106 to 108: “Oviposition traps are used to collect populations of Ae. aegypti and Ae. albopictus as tools for mosquito surveillance in different studies in Brazil, since this method has high sensitivity in detecting the presence and comparing infestations in different areas”

- Lines 98 to 103: this is a very big sentence. It could be divided into two or three small phrases.

Authors: Done. This sentence was changed as required on lines 112 to 115 (first phrase) and on lines 116 to 118 (second phrase).

3. Results

- Line 158: "were collected" appeared two times in this sentence.

Authors: Done. It was removed "were collected" on line 172.

- Lines 179 to 182: I think it is Figure 3A and not 2B; the authors did not cite Figure 3B.

Authors: Done. The Figure 2B was changed to 3A on line193. The Figure 3B was mentioned on line 196.

- Lines 184 and 185: I think it is Figure 3C and not A.

Authors: Done. The Figure 3A was changed to 3C on line 198.

- Lines 187 and 188 - Figures 3A and B: These maps (A and B) also represent the spatial distribution, not only the temporal one.

Authors: Done. The Figures (A and B) represent spatial and temporal distribution. The information was put on lines 200 and 202.

- line 196 - Figure 4: This figure, in my point of view, is not necessary, the authors could comment on the information about dengue in the text and insert neighbor municipalities of Morretes in figure 1.

Authors: The authors apologize your suggestion, but we think to this figure clarify the information about dengue cases no coast the state of Paraná. In this sense, we decided left it.

Reviewer 2 Report

The article entitled “Spatial and temporal distribution of Aedes aegypti and Aedes 2 albopictus oviposition in a recent area of dengue virus trans-3 mission” has been intended to express a relevant subject in the current scenario. However, many of the portions lack sufficient and recent perspectives from international and national studies. Moreover, the authors should check the whole manuscript for grammatical errors. In addition to the major comments attached below, I recommend the article for Major Revision and the article cannot be accepted for this journal in this form. Once the authors have rectified the concerns raised by me, it can be accepted for publication.

Title

The title should be modified according to your objective/focus/ ]current study area. Consider the following part in the title; “in a recent area of dengue virus transmission”, authors should provide a precise idea of the study area in the title part, whether it is worldwide or it is a specific area of the country, or a specific to a particular country.  I think it is “Paraná 28 State, Brazil” if so include the same in the title part.

Abstract

L27-36

Add a background of the subject in the abstract part. It is preferable to add the core content of the script in the abstract part. The abstract was found very small, if it is based on the journal guidelines, then no changes are required.  

Introduction

Try to add more studies/ information concerning the spatial and temporal distribution in the introduction part.

Add the recent advancements in the same perspective in the introduction part.

Add the perspectives of recent studies focusing on the same concept over the globe.

Should add a research gap in the introduction part.

Also, add how relevant the current topic/subject.

L 41-43

The second sentence “Arthropod-borne viruses, such as 42 dengue (DENV), Zika (ZIKV), and yellow fever (YFV) chikungunya virus (CHIKV) [1–3].” In the introduction, the part should be modified. Try to link the first and second sentences or merge both of them if possible.

L 44-47

The authors should specify the study  area for the studies that are used for supporting the L 44-47

 Statements.

L 48-49

Rewrite the sentence and modify the same for author-friendly nature. 

L 57-59

Add more points regarding the major contributing parameters in this regard.

L 59-61. Which temperature? The authors should maintain scientific consistency throughout the manuscript.  Hence, for the major sentence in the manuscript, especially allied with environment, habitat, and diseases, the authors should add precise data support for the same. In this case, here the authors should update the sentence with the following points.

1, Major contributing facts

2. Which seasons

3. major locations

Materials and Methods

L89-92 Rewrite the sentence

L95-97 This is actually methodology section. Not an introduction part or not discussion. You can prefer the methodology suggested or used by the previous investigation. However, it is not necessary to explain these technologies or methods are for this… etc etc .In this regard, it is mandatory to rewrite this part.

L 102 -103

From where you have got the infection data, if it is from the hospital, here you have the space to mention the same. You can also merge the sentence from L112-114.

L144

Add reference for Poisson Generalized Linear Mixed Model (GLMM) followed by variable selection 144 through likelihood ratio (LRT) test.

L 150

Which are the other climate variable? You have to mention the same. Otherwise, you can delete the same from here. The unclear sentence makes the readers more confused, hence the authors should consider this comment throughout the manuscript.  

Result

L 170-171

Check for grammatical errors and Update the legend.

Figure 2. Frequencies of Aedes aegypti and Aedes albopictus adults by sex identified from 170 eggs in Morretes, Paraná, period of October 2017 to September 2018.

Figure 3

It is better to add a border for A,B, and C or just provide enough space to separate the ABC image.

Discussion

It should be updated with recent studies.

L261. After this line provide strong recent studies (2022) to support your findings.

L 263-L 266

Should be expanded with recent studies

L 279-281

Rewrite sentences with recent studies. The authors have mentioned that the articles in 2009 as recent, It is strictly noted to be critical. so updated with recent studies and rewrite the same. In addition compare your data from an international perspective, since researchers from all over the world are sharing and interpreting their data frequently.

Write a conclusion for the article.

Author Response

We authors thank the reviewers for their comments and suggestions to improve our manuscript. We have carefully revised the manuscript and considered all modifications as suggested. Please, find below point-by-point responses to reviewer as requested, indicating which changes have been made, and where they have been included in the manuscript. Furthermore, the manuscript was revised by ECB - English Consulting Brazil.

Answers to Reviewer

Reviewer #2: The article entitled “Spatial and temporal distribution of Aedes aegypti and Aedes 2 albopictus oviposition in a recent area of dengue virus trans-3 mission” has been intended to express a relevant subject in the current scenario. However, many of the portions lack sufficient and recent perspectives from international and national studies. Moreover, the authors should check the whole manuscript for grammatical errors. In addition to the major comments attached below, I recommend the article for Major Revision and the article cannot be accepted for this journal in this form. Once the authors have rectified the concerns raised by me, it can be accepted for publication.

1. Title

The title should be modified according to your objective/focus/current study area. Consider the following part in the title; “in a recent area of dengue virus transmission”, authors should provide a precise idea of the study area in the title part, whether it is worldwide or it is a specific area of the country, or a specific to a particular country.  I think it is “Paraná 28 State, Brazil” if so include the same in the title part.

Authors: We agree and the suggestion was accepted. The title was changed to “Spatial and temporal distribution of Aedes aegypti and Aedes albopictus oviposition on the coast of Paraná, Brazil, a recent area of dengue virus transmission”

2. Abstract

- L27-36 - Add a background of the subject in the abstract part. It is preferable to add the core content of the script in the abstract part. The abstract was found very small, if it is based on the journal guidelines, then no changes are required. 

Authors: Thank you very much for this suggestion. The journal´s guidelines require a single paragraph of about 200 words at most, so in this sense it was possible to include: “Aedes aegypti and Aedes albopictus are considered the most important vectors of arboviruses in the world. Aedes aegypti is the primary vector of dengue, urban yellow fever, chikungunya and zika in Brazil, and Ae. albopictus is considered a potential vector”. 

3.  Introduction

Try to add more studies/ information concerning the spatial and temporal distribution in the introduction part. Add the recent advancements in the same perspective in the introduction part. Add the perspectives of recent studies focusing on the same concept over the globe. Should add a research gap in the introduction part. Also, add how relevant the current topic/subject.

Authors: We added studies published in 2019 (references 7, 9, 51, 62, 72), 2020 (references 44, 52, 56, 66, 70, 73), 2021 (references 30, 41, 45, 64, 74) and 2022 (references 46, 60, 71).  In addition, to further support our question in the manuscript, we included the paragraph: “The Paraná State Health Department (SESA/PR) reported the first important dengue epidemic in the city of Paranaguá in 2016, as well as the first occurrence of ZIKV and CHIKV infections [14]. Paranaguá and Antonina are the main city ports along the coast of the State of Paraná, Brazil. All cargoes which leave the port of Antonina are routed through Morretes, which is also an important tourist city. Aedes albopictus was the predominant species in the area until 2015, when the first autochthonous cases of DENV and the detection of Ae. aegypti were reported.” on lines 81-87.

L 41-43 The second sentence “Arthropod-borne viruses, such as 42 dengue (DENV), Zika (ZIKV), and yellow fever (YFV) chikungunya virus (CHIKV) [1–3].” In the introduction, the part should be modified. Try to link the first and second sentences or merge both of them if possible.

Authors: Done. The sentence was changed to Aedes aegypti and Aedes albopictus are sympatric, cosmopolitan and epidemiologically important species involved in transmitting arboviruses such as dengue (DENV), zika (ZIKV), yellow fever (YFV), and chikungunya viruses (CHIKV)” on lines 46 to 48.

L 44-47 The authors should specify the study area for the studies that are used for supporting the L 44-47

Authors: The suggestion was accepted. The authors added the locality: “These species have similar geographical distribution and share microhabitats in some regions such as Florida and Rio de Janeiro, including artificial vessels containing water” on line 48-50.

 Statements.

L 48-49 Rewrite the sentence and modify the same for author-friendly nature. 

Authors: Thanks. The suggestion was accepted and modified to: “Studies carried out on mapping the global distribution of Ae. aegypti and Ae. albopictus point to an expansion in the range of habitat suitability for these species and predict that their dissemination will occur despite environmental changes, since the species are adapting to anthropogenic ecological niches using spatial dispersion” on lines 54-57.

L 57-59 Add more points regarding the major contributing parameters in this regard.

L 59-61. Which temperature? The authors should maintain scientific consistency throughout the manuscript.  Hence, for the major sentence in the manuscript, especially allied with environment, habitat, and diseases, the authors should add precise data support for the same. In this case, here the authors should update the sentence with the following points.

1. Major contributing facts

2. Which seasons

3. major locations

Authors: Done. The sentence was changed “Aedes aegypti and Ae. albopictus occur in tropical and subtropical regions as such environmental characteristics allow reproduction and provide adequate habitat, and most of Brazil is within this climatic zone [8]. The southern region of Brazil differs from other Brazilian regions due to a distinction between seasons and also by the thermal amplitude; it is the coldest region in the country, with snow in some locations during the winter [18]. The coast of Paraná has a humid subtropical climate (mesothermal) according to the Koppen classification [19–21], with the average temperature in the cold months being below 18ºC and an average temperature above 22ºC in the summers, concentrating the rains in the summer, despite not having a defined dry season [22–24], on lines 65-73.

4. Materials and Methods

L89-92 Rewrite the sentence

Authors: Done. “Oviposition traps are used to collect populations of Ae. aegypti and Ae. Albopictus as tools for mosquito surveillance in different studies in Brazil, since this method has high sensitivity in detecting the presence and comparing infestations in different areas.” Lines 106-108.

L95-97 This is actually methodology section. Not an introduction part or not discussion. You can prefer the methodology suggested or used by the previous investigation. However, it is not necessary to explain these technologies or methods are for this… etc etc. In this regard, it is mandatory to rewrite this part.

Authors: We agree and the phrase was removed. The information was rewrite on the lines 106-108.

L 102 -103 From where you have got the infection data, if it is from the hospital, here you have the space to mention the same. You can also merge the sentence from L112-114.

Authors: Done. The information about infection data was obtained from Paraná State Health Department (SESA/PR). The reference was put on lines 116-120: “The monthly collection was from September 2017 to July September 2018, one year after the first notification of an autochthonous cases of dengue in the municipality of Paranaguá by the Paraná State Health Department (SESA/PR). The data on autochthonous cases of dengue fever in the city of Morretes and coastal municipalities from 2014 until 2020 were obtained by SESA”.

We accepted the suggestion about “You can also merge the sentence from L112-114”. The phrase is on lines 118-120: “The data on autochthonous cases of dengue fever in the city of Morretes, and coastal municipalities from 2014 until 2020 were obtained by SESA”.

L144 Add reference for Poisson Generalized Linear Mixed Model (GLMM) followed by variable selection 144 through likelihood ratio (LRT) test.

Authors: Done. The reference was put on line 159.

L 150 Which are the other climate variable? You have to mention the same. Otherwise, you can delete the same from here. The unclear sentence makes the readers more confused, hence the authors should consider this comment throughout the manuscript.  

Authors: Thank you for pointing this out. All climatic variables were already stated in the text, so we rephrased the sentence to improve clarity. "All other available climate variables, i.e. maximum temperature, minimum temperature, and their combination were excluded due to collinearity (r > 0.7)" on line 164-165.

5. Result

L 170-17 Check for grammatical errors and Update the legend.

Figure 2. Frequencies of Aedes aegypti and Aedes albopictus adults by sex identified from eggs in Morretes, Paraná, period of October 2017 to September 2018.

Authors: Done. The legend changed to: “Frequencies of Aedes aegypti and Aedes albopictus adults by sex identified from hatching eggs in Morretes, Paraná, period of October 2017 to September 2018” on line 184-185.

Figure 3 It is better to add a border for A,B, and C or just provide enough space to separate the ABC image.

Authors: Done. In fact, the layout is better.

6. Discussion

It should be updated with recent studies.

Authors: We added studies published in 2019 (references 7, 9, 51, 62, 72), 2020 (references 44, 52, 56, 66, 70, 73), 2021 (references 30, 41, 45, 64, 74) and 2022 (references 46, 60, 71).

L261 After this line provide strong recent studies (2022) to support your findings.

Authors: We added studies published in 2019 (references 7, 9, 51, 62, 72), 2020 (references 44, 52, 56, 66, 70, 73), 2021 (references 30, 41, 45, 64, 74) and 2022 (references 46, 60, 71).

L 263-L 266 Should be expanded with recent studies

Authors: We added studies published in 2019 (references 7, 9, 51, 62, 72), 2020 (references 44, 52, 56, 66, 70, 73), 2021 (references 30, 41, 45, 64, 74) and 2022 (references 46, 60, 71). 

L 279-281 Rewrite sentences with recent studies. The authors have mentioned that the articles in 2009 as recent, It is strictly noted to be critical. so updated with recent studies and rewrite the same. In addition compare your data from an international perspective, since researchers from all over the world are sharing and interpreting their data frequently.

Authors: We added studies published in 2019 (references 7, 9, 51, 62, 72), 2020 (references 44, 52, 56, 66, 70, 73), 2021 (references 30, 41, 45, 64, 74) and 2022 (references 46, 60, 71). However, some older references such as 3, 15-17, 21-22, 71-72, bring citations of notifications, analysis methods and information to support the discussion of the findings with contemporary studies.

Write a conclusion for the article.

Authors: The conclusion was added on lines 347-354.

Round 2

Reviewer 2 Report

They have addressed my comments. Hence the article can be accepted for Publication in this form in Tropical Medicine and Infectious Disease.